# EXTREME COMBINED COMPRESSION OF LARGE LANGUAGE MODELS THROUGH JOINT OPTIMIZATION

## ABSTRACT

Post-Training Quantization (PTQ) and Sparsification (PTS) are dominant methods in the compression of Large Language Models (LLMs) due to their minimal resource usage and generalizability. It is a natural idea to integrate quantization and sparsification in a unified framework, which however, often results in substantial accuracy losses. Here we argue that, the key lies in optimization. This paper introduces a novel joint optimization strategy that concurrently mitigates errors induced by both sparsification and quantization. Unlike sequential approaches, our method employs learnable transformation matrices to simultaneously optimize errors across both dimensions, preventing the typical misalignments associated with sequential optimizations. Furthermore, we present a reordering mechanism within the learnable mask sparsification process to maintain consistent sparsity ratios. This mechanism ensures the prioritization of the least important weights during each update iteration, thus enhancing the stability of the compression process. Our approach demonstrates considerable performance enhancements across diverse models and datasets, with the most notable gains observed under conditions of extremely low-bit quantization and high sparsity ratios. For example, in the LLaMA2-13b model with weight quantization at 2 bit and a $75\%$ sparsity configuration, our method surpasses the state-of-the-art (SOTA) by $9.03\%$ in average accuracy across five zero-shot tasks. Meanwhile, in the newest LLaMA3-8b model, with weight quantization at 3 bit and a $50\%$ sparsity configuration, our method outperforms the SOTA by $4.58\%$ ($56.86\%$ vs $52.28\%$) in zero-shot tasks and achieves a perplexity reduction of $4.45$ on the WikiText2 dataset ($10.78$ vs $15.23$).

## 1 INTRODUCTION

The remarkable capabilities of large language models (LLMs) (Touvron et al., 2023a;b; Brown et al., 2020; Le Scao et al., 2023; Zhang et al., 2022) have garnered considerable interest across a multitude of disciplines. Yet, the escalating dimensions of these models present significant deployment challenges on consumer-grade graphics processing units (GPUs). As a result, the topics of model compression and acceleration have increasingly become focal points within the research community. Two prevalent techniques—model quantization and sparsification—are recognized for their effectiveness in both compressing and accelerating model inference processes. Quantization (Frantar et al., 2023; Yuan et al., 2023; Lin et al., 2023; Xiao et al., 2023; Shao et al., 2023; Ma et al., 2024) involves mapping high-precision floating-point parameters to lower-bit fixed-point representations. This method not only reduces the model size by minimizing the bit requirements but also boosts memory access efficiency, a crucial factor in speeding up computation. Especially on hardware platforms equipped to accelerate bitwise operations, or on processors specifically optimized for low-bit computations, quantized models can achieve notably faster processing speeds. Additionally, model sparsification (Frantar & Alistarh, 2023; Sun et al., 2023; Zhang et al., 2023; Xu et al., 2024) compresses models by transforming dense weight matrices into sparse formats through the application of sparse masks. This approach can significantly quicken computation on hardware platforms that are optimized for handling sparse matrix operations, thereby enhancing the overall execution speed of the models.

While model compression techniques such as quantization and sparsification accelerate computational processes, they invariably entail some degree of performance degradation. The extensive

fine-tuning required for large language models, which often comprise hundreds of millions of parameters, involves significant time and computational resources. As a result, compression methods that bypass extensive training phases are gaining prominence. Among these, post-training quantization, which necessitates no fine-tuning or only minor calibration, has been widely adopted from the era of convolutional neural networks through to that of large language models. GPTQ (Frantar et al., 2023) exemplifies advanced quantization techniques, implementing progressive quantization with the Optimal Brain Surgeon (OBS) (Hassibi et al., 1993) algorithm to mitigate quantization errors by revising high-precision parameters. Similarly, AWQ (Lin et al., 2023) performs input channel scaling, utilizing activation statistics to achieve equivalent scaling of weight activations, effectively managing the impact of outliers on quantization. OmniQuant (Shao et al., 2023) pioneered the adaptation of scaling and clipping parameters into learnable entities, facilitating block-wise optimization using small data batches. In the realm of large language models, non-training-dependent sparsification methods are increasingly favored over those requiring fine-tuning. For instance, Wanda (Sun et al., 2023) posits that activations must be included in weight importance assessments, suggesting that a singular weight statistic is insufficient to capture a weight's true role within the model. Dsnot (Zhang et al., 2023) introduces a training-free approach by iteratively updating sparse masks, dynamically reducing the density of weights. Moreover, BESA (Xu et al., 2024) innovates by applying a learnable factor to a set of candidate sparsity rates, assigning variable sparsity to different linear layers, and enabling dynamic mask optimization through adaptive sparsity.

The two compression techniques under consideration—model quantization and sparsification—do not require a training phase, which allows for their combination to potentially enhance model compression further. However, this amalgamation significantly impairs the model's performance. Even with advancements in cutting-edge compression algorithms, the performance of the model severely deteriorates under configurations characterized by low bit-width and high compression ratios. A primary contributor to this decline is the sequential implementation of the quantization and sparsity algorithms. Our experiments indicate that initiating with quantization optimization followed by the application of weight sparsity amplifies the quantization errors, consequently increasing the overall mean squared error loss.

This paper introduces a joint optimization strategy designed to concurrently minimize errors associated with quantization and sparsity. More precisely, following each iteration that establishes a sparsity mask, we calibrate the quantization process using small data batches, applying the sparsity mask to the weights concurrently. Losses resulting from quantization and sparsity are optimized using a learnable transformation matrix, which adjusts dynamically. This joint optimization approach is designed to be orthogonal to a broad spectrum of existing quantization and sparsity algorithms, and it has demonstrated superior performance across a variety of algorithmic configurations. Furthermore, because the learnable transformation matrix modifies the distribution of weight data in each iteration, the learnable mask sparsity strategy must consistently ensure that weights with lower importance factors are preferentially masked. To maintain this selection, we implement a reordering of the weights based on sparsity metrics at every iteration, prioritizing the least important weights to sparsity. This reordering strategy promotes stable convergence and maintains optimal performance throughout the training process. Ultimately, our method excels in the combined compression of large language models, particularly under conditions of low-bit and high-sparsity configurations. In conclusion, the key contributions of this study are summarized as follows:

- We introduce a novel joint optimization strategy that integrates compression techniques to concurrently minimize errors associated with quantization and sparsity. This methodology is designed to be orthogonal to a wide spectrum of existing compression algorithms, demonstrating unparalleled performance across diverse algorithmic combinations.

- Furthermore, we have proposed a dynamic reordering method to enhance the efficacy of learnable masks. By systematically reordering weight rows according to sparsity metrics at each iteration, this method ensures the consistent masking of the least significant weights. This process not only stabilizes the convergence of the optimization algorithm but also maximizes the effectiveness of learnable masking, contributing to more reliable model training outcomes.

- Our techniques have set new benchmarks in the field of combined compression for large language models, particularly in configurations characterized by low bit-width and high sparsity. Notably, our strategy improved average performance by 9.03% over traditional

sequential compression methods in five zero-shot tasks on the LLaMA2-13b model configured with 2-bit quantization and 75% sparsity. Additionally, in a setting with 3-bit quantization and 50% sparsity on the LLaMA3-8b model, our method surpassed the current state-of-the-art by 4.58% (achieving 56.86% compared to 52.28%) in zero-shot tasks, and significantly reduced perplexity by 4.45 on the WikiText2 dataset (10.78 compared to 15.23).

## 2 RELATED WORK

The substantial parameter count of large language models presents considerable challenges for the application of quantization and sparsity methods, especially when fine-tuning is required. Given these challenges, this paper focuses on the efficient strategies of post-training quantization and sparsity.

**Post-training Quantization (PTQ).** In the convolutional neural network (CNN) domain, several post-training quantization methods, such as Adaround (Nagel et al., 2020), BRECQ (Li et al., 2021), and QDROP (Wei et al., 2022a), employ adaptive rounding techniques for model weights. This involves reconstructing the model either layer-wise or block-wise to optimize the rounding parameters, thereby minimizing quantization errors. Specifically, in large language models, outliers significantly complicate the quantization process. GPTQ (Frantar et al., 2023) addresses this by correcting full-precision parameters using the OBS (Hassibi et al., 1993) algorithm. Furthermore, numerous studies (Wei et al., 2022b; Lin et al., 2023; Xiao et al., 2023; Yuan et al., 2023; Shao et al., 2023; Ma et al., 2024) have adopted equivalent transformation operations—including scaling, shifting, rotating, and rearranging—to relocate the effects of activation outliers to the weights, effectively diminishing their adverse impacts. Furthermore, the quantization method based on orthogonal matrix (Ashkboos et al., 2024) transformation efficiently generates a quantized model without the need for retraining. This orthogonally constrained sampling matrix effectively mitigates the impact of outliers in activations on the quantization process.

**Post-training Sparsity.** Similarly, outliers in activations detrimentally influence sparsity performance within large language models. Traditional pruning algorithms can severely degrade model performance, typically necessitating subsequent fine-tuning for performance recovery. To address this, a variety of post-training sparsity techniques (Frantar & Alistarh, 2023; Sun et al., 2023; Zhang et al., 2023; Xu et al., 2024) have been developed to efficiently implement sparsity in large models. They range from employing single-dimensional to multi-dimensional importance metrics, from those based on statistical characteristics to those relying on gradient-based optimization, and evolve from static to dynamic sparsity approaches. The evolution of post-training sparsity methods has significantly enhanced the adaptability of large language models across diverse computing platforms, promoting broader deployment and application.

## 3 METHODOLOGY

In this section, we initially introduce foundational concepts and establish notation conventions relevant to quantization and sparsification. We then propose the application of a joint optimization combination compression method in large language models. This method, in contrast to other combined compression strategies, consistently achieves stable and low mean squared error losses throughout the optimization process. Finally, to ensure effective sparsification of weights with low importance, we introduce a reordering strategy that maintains the priority of non-critical weights, thus preserving the model's overall performance.

### 3.1 PRELIMINARY

**Quantization.** Model quantization typically involves mapping the weights or activation parameters to low-bit integers to compress the model. To clarify, we define our quantization function as follows:

$$\mathcal{Q}(x) = \Delta * \left( clamp \left( \left\lfloor \frac{x}{\Delta} \right\rceil + zp, 0, 2^n - 1 \right) - zp \right). \tag{1}$$

Where $\Delta$ is the quantization step size, $zp$ is the zero point, $n$ is the quantization bit width, and the $clamp(\cdot)$ operation truncates data that exceeds the range to the upper or lower bounds. The quantization step size is commonly used to map the distribution range of the input data $x$ to the range of the quantized fixed-point distribution. Consequently, calculating the step size factor and the zero point based on the maximum and minimum values of $x$ is a standard approach, as follows:

$$\Delta = \frac{max(x) - min(x)}{2^n - 1}, \quad zp = \left\lfloor \frac{-min(x) * (2^n - 1)}{max(x) - min(x)} \right\rceil . \tag{2}$$

The presence of outliers in large language models significantly impairs performance (Frantar et al., 2023; Lin et al., 2023; Xiao et al., 2023) when the maximum and minimum values are used to determine the quantization step size. These outliers compress the bulk of non-outlier values towards near-zero quantized integers, introducing substantial rounding errors for a majority of parameters. Additionally, directly truncating outliers can lead to a marked decline in performance (Bondarenko et al., 2024). To mitigate these issues, the maximum and minimum values are often scaled by a learnable factor (Shao et al., 2023), enabling the determination of an optimal truncation range via gradient descent. This approach is formalized as follows:

$$\Delta = \frac{\alpha * max(x) - \beta * min(x)}{2^n - 1}, \quad zp = \left\lfloor \frac{-\beta * min(x) * (2^n - 1)}{\alpha * max(x) - \beta * min(x)} \right\rceil . \tag{3}$$

Where $\alpha$ and $\beta$ are learnable parameters, adjusted with a small sample of data through gradient-based calibration.

**Sparsification.** Sparsification of LLMs entails converting dense weight matrices into sparse matrices through the application of a sparse mask. During this process, our objective is to minimize the mean squared loss between the features before and after sparsification. The sparsity rate $p$ is defined as the ratio of the number of sparse elements to the total number of weight parameters. Specifically, for a given weight matrix $W$ and activations $X$, the sparsification optimization problem is formalized as follows:

$$\arg\min_{M} \|XW - X(M \odot W)\|_F^2, \quad p = 1 - \frac{\|W\|_0}{C_{in} * C_{out}}. \tag{4}$$

Where $M$ denotes the sparse mask, $\odot$ represents the Hadamard product, $\|W\|_0$ indicates the count of non-zero elements in $W$, and $C_{in}/C_{out}$ correspond to the number of input and output channels, respectively, in the weight matrix $W$.

**Blockwise Optimization.** Due to the large number of parameters in LLMs, in order to save the memory used by the model, we adopt a block-wise optimization strategy. Specifically, for the optimization process of quantization or sparsification, our optimization problem is constructed as follows:

$$\arg\min_{W'} \|f_i(X, W) - f_i(X, W')\|_F^2 . \tag{5}$$

In this formulation, for quantization and sparsification respectively, $W' = \mathcal{Q}(W)$ or $W' = M \odot W$ is used. The term $f_i$ denotes the $i$-th transformer block.

## 3.2 JOINT OPTIMIZATION

Recent methodologies (Xu et al., 2024) in combined compression typically sequence quantization and sparsification processes. Given the prevalence of outliers in LLMs, contemporary quantization techniques (Shao et al., 2023; Ma et al., 2024) employ learnable scales or matrices to execute equivalent transformations on both weights and activation vectors. Consider a scenario where the input activations are denoted as $X$ and the weights as $W$, the quantization optimization problem can be formalized as follows:

$$\arg\min_{A} \|XW - XA^{-1}\mathcal{Q}(AW)\|_F^2 . \tag{6}$$

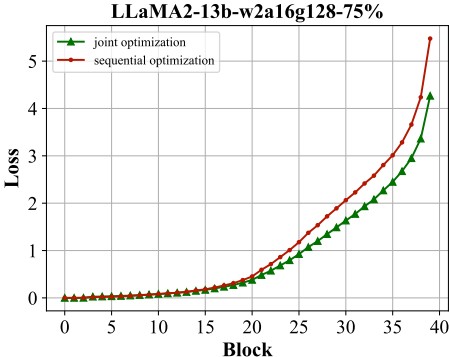
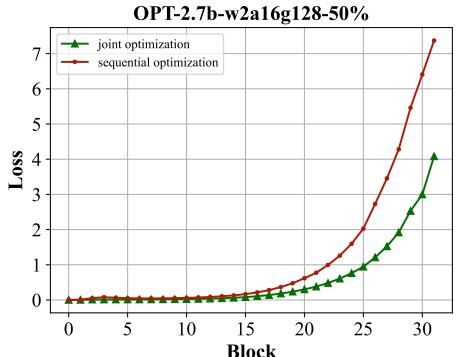

Figure 1: Comparison of Mean Squared Error (MSE) losses for combined compression techniques using joint and sequential optimization methods. The label "w2a16g128" indicates weight-only quantization at a 2-bit and a group size of 128. The "75%" refers to the sparsity rate implemented. "Block" denotes transformer blocks of various depths. MSE losses of features are computed after 20 epochs of optimization across each block. The sparsification is performed using the Wanda (Sun et al., 2023) method, and quantization is conducted with the OmniQuant (Shao et al., 2023) method.

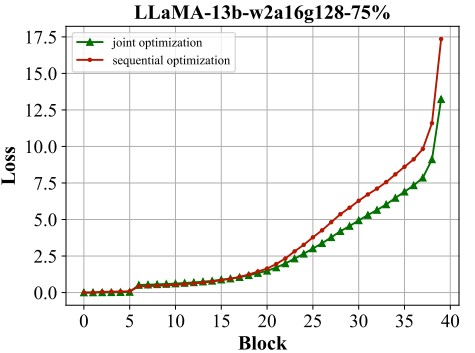
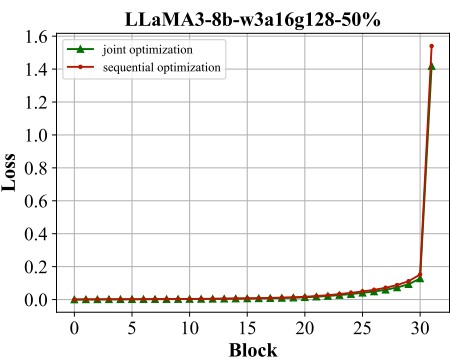

Figure 2: Comparison of Mean Squared Error (MSE) losses for combined compression techniques using joint optimization and sequential optimization methods on LLaMA1&3.

Where $A$ is a learnable transformation matrix. The latest combined compression methods (Xu et al., 2024) first optimize quantization errors according to Eq. 6, and subsequently calculate the importance factors for the quantized weights $\mathcal{Q}(AW)$ to determine the sparse mask. Specifically, this process can be described as follows: assuming the matrix $A_{opt}$ makes $\left\| XW - XA_{opt}^{-1}\mathcal{Q}(A_{opt}W) \right\|_F^2$ optimal, the sparse mask $M$ is then optimized to minimize $\left\| XW - XA_{opt}^{-1}\left(M \odot \mathcal{Q}(A_{opt}W)\right) \right\|_F^2$. This sequential optimization approach results in quantization optimization that does not account for the loss introduced by sparsification. Consequently, this two-stage combined compression method causes the matrix $A$ and the mask $M$ to achieve only local optimality in their respective stages. As a result, the overall mean squared error loss of the features is difficult to minimize globally. Therefore, we propose the following combined compression method based on joint optimization:

$$\underset{A,M}{\arg\min} \left\| XW - XA^{-1}\left(M \odot \mathcal{Q}(AW)\right) \right\|_F^2. \tag{7}$$

Specifically, in cases where the sparsification method statically determines the mask without relying on gradient information, the matrix $M$ is computed before the optimization process begins and remains unchanged. Throughout the optimization process, the joint optimization method consistently demonstrates superior performance in reducing mean squared error loss across various models. As illustrated in Fig. 1, for the LLaMA2-13b and OPT-2.7b models, joint optimization consistently

achieves lower loss compared to sequential optimization across different transformer blocks, under varying bit widths and sparsity rates.

Additionally, we conduct experiments to analyze the distribution of mean squared error (MSE) losses on LLaMA1&3. As illustrated in Fig. 2, the MSE losses resulting from joint optimization are consistently lower than those from sequential optimization across both LLaMA1&3, highlighting the effectiveness of the joint optimization approach.

For large language models, the objective of block-wise combined compression optimization is defined as follows:

$$\arg \min_{A,M,\delta} \left\| f_i(X, W) - f_i((X - \delta)A^{-1}, M \odot \mathcal{Q}(AW), b + \delta W) \right\|_F^2 . \tag{8}$$

Where $\delta$ represents a learnable offset designed to align the distributions across different channels of weights and activations. Consistent with previous research (Shao et al., 2023; Ma et al., 2024; Xu et al., 2024), we overlook the impact of quantization and sparsification on this offset when compensating for the bias in linear layers. Detailed derivations are provided in the Appendix A.1.

### 3.3 REORDERING

To reduce the complexity of optimization, the learnable mask sparsification strategy typically establishes a set of candidate sparsity rates for each row of the weight matrix $W$. Each candidate sparsity rate is multiplied by a learnable factor to ascertain the optimal sparsity rate for every row. Concurrently, the rows of $W$ are organized according to a sparsity importance factor, and weights deemed less important are sparsified based on the aggregate sparsity rate derived from the candidate rates. However, the learnable transformation matrix $A$, as specified in Eq. 7, alters the distribution of weights with each iteration. Consequently, the importance of weights changes dynamically throughout the optimization process. We propose that at the beginning of each optimization iteration, the quantized weights $\mathcal{Q}(AW)$ are reordered based on the following importance metric:

$$\gamma_{i,j} = \|X_{:,i}\|_F^2 \cdot |W_{i,j}| . \tag{9}$$

By employing a reordering strategy, the dynamic mask sparsification method consistently and accurately targets weights with lower importance in each iteration. This approach significantly enhances the generalizability and effectiveness of the joint optimization method, thereby ensuring consistently superior performance across various models and datasets.

## 4 EXPERIMENTS

### 4.1 SETTINGS

**Implementation Details.** The implementation details of our proposed joint optimization strategy are outlined as follows. Consistent with established quantization (Shao et al., 2023; Ma et al., 2024) and sparsification (Frantar & Alistarh, 2023; Sun et al., 2023; Zhang et al., 2023; Xu et al., 2024) methodologies, our calibration dataset comprises 128 segments, each consisting of 2048 tokens sampled from the WikiText2 (Merity et al., 2016) training corpus. In terms of hyperparameter configuration, when employing AffineQuant (Ma et al., 2024) for quantization, we set the stability factor $\alpha = 0.1$. For all experiments utilizing DSnoT (Zhang et al., 2023) as the sparsification method, we define the maximum cycle $T = 50$ and the update threshold $\epsilon = 0.1$. In experiments involving BESA (Xu et al., 2024) as the sparsification approach, the sparsity radio $\beta$ is set to $5e^0$. Our proposed methodology is implemented in PyTorch (Paszke et al., 2019) and leverage the HuggingFace Transformers library Wolf et al. (2019) for data and model management.

**Baselines.** We apply joint optimization strategy to the LLaMA (Touvron et al., 2023a), LLaMA2 (Touvron et al., 2023b), LLaMA-3, and OPT (Zhang et al., 2022) families, representing the forefront of open-source Large Language Models (LLMs). Notably, our approach transcends limitations in model size, enabling optimization across a spectrum ranging from 125 million to 70 billion parameters on a single NVIDIA A800 GPU with 80GB of memory. To benchmark our methodology, we juxtapose it with the state-of-the-art (SOTA) sequential optimization proposed by

Table 1: Comparison of perplexity and zero-shot dataset accuracy between sequential optimization and joint optimization at 75% sparsity on LLAMA1&2. "Sequential" and "Joint" denote the sequential and joint combined compression methods, respectively. "w2a16g128" refers to 2-bit weight-only quantization with a group size of 128. "Affine" indicates that quantization is performed using AffineQuant (Ma et al., 2024).

| Model | Bits | Sparsity | Method | PPL ↓ | | | Accuracy (%) ↑ | | | | | |
| | | | | WikiText2 | C4 | Avg. | BoolQ | HellaSwag | WinoGrande | ARC-c | ARC-e | Avg. |
|---|---|---|---|---|---|---|---|---|---|---|---|---|
| LLaMA2-13B | w16a16 | − | − | 4.88 | 6.73 | 5.81 | 80.55 | 60.04 | 72.21 | 48.46 | 79.37 | 68.13 |
| | w2a16g128 | 75% | Sequential-Wanda | 2677.90 | 3837.04 | 3257.47 | 37.82 | 26.19 | 47.35 | 22.18 | 25.33 | 31.77 |
| | w2a16g128 | 75% | Joint-Wanda | 83.46 | 123.17 | 103.32 | 57.18 | 29.01 | 49.25 | 20.64 | 32.36 | 37.69 |
| | w2a16g128 | 75% | Joint-Wanda-Affine | 26.86 | 38.82 | **32.84** | 62.23 | 31.19 | 51.46 | 22.01 | 37.12 | **40.80** |
| | w2a16g128 | 75% | Sequential-DSnoT | 3918.30 | 7890.29 | 5904.30 | 37.82 | 26.23 | 48.38 | 20.81 | 25.46 | 31.74 |
| | w2a16g128 | 75% | Joint-DSnoT | 112.16 | 139.08 | 125.62 | 51.34 | 28.60 | 49.17 | 20.39 | 32.11 | 36.32 |
| | w2a16g128 | 75% | Joint-DSnoT-Affine | 59.92 | 76.93 | **68.43** | 52.53 | 29.19 | 51.30 | 21.24 | 31.48 | **37.15** |
| LLaMA-7B | w16a16 | − | − | 5.63 | 7.07 | 6.35 | 75.10 | 56.95 | 69.85 | 41.89 | 75.29 | 63.82 |
| | w2a16g128 | 75% | Sequential-Wanda | 3521.03 | 3567.52 | 3544.28 | 37.82 | 25.72 | 48.69 | 21.33 | 26.17 | 31.95 |
| | w2a16g128 | 75% | Joint-Wanda | 1247.68 | 1398.08 | 1322.88 | 37.82 | 25.86 | 49.56 | 21.24 | 26.76 | 32.25 |
| | w2a16g128 | 75% | Joint-Wanda-Affine | 44.79 | 70.32 | **57.56** | 50.12 | 28.22 | 49.80 | 20.22 | 32.70 | **36.21** |
| | w2a16g128 | 75% | Sequential-DSnoT | 3146.59 | 4742.01 | 3944.30 | 37.82 | 25.99 | 50.43 | 21.67 | 26.09 | 32.40 |
| | w2a16g128 | 75% | Joint-DSnoT | 1297.72 | 1468.20 | 1382.96 | 37.82 | 25.72 | 50.90 | 20.05 | 27.27 | 32.35 |
| | w2a16g128 | 75% | Joint-DSnoT-Affine | 85.69 | 126.70 | **106.20** | 39.41 | 27.50 | 52.80 | 20.98 | 35.01 | **35.14** |
| LLaMA-13B | w16a16 | − | − | 5.03 | 6.61 | 5.82 | 77.98 | 59.91 | 72.77 | 46.50 | 77.35 | 66.90 |
| | w2a16g128 | 75% | Sequential-Wanda | 1345.77 | 1705.19 | 1525.48 | 37.82 | 26.10 | 50.03 | 20.90 | 25.67 | 32.10 |
| | w2a16g128 | 75% | Joint-Wanda | 61.57 | 73.72 | 67.65 | 45.87 | 29.55 | 52.40 | 21.58 | 34.13 | 36.71 |
| | w2a16g128 | 75% | Joint-Wanda-Affine | 30.50 | 46.90 | **38.70** | 56.85 | 30.76 | 54.22 | 20.22 | 35.81 | **39.57** |
| | w2a16g128 | 75% | Sequential-DSnoT | 1408.23 | 1913.72 | 1660.98 | 37.82 | 26.00 | 48.85 | 21.24 | 26.38 | 32.06 |
| | w2a16g128 | 75% | Joint-DSnoT | 95.21 | 93.78 | 94.50 | 47.43 | 29.31 | 50.74 | 20.05 | 32.32 | 35.97 |
| | w2a16g128 | 75% | Joint-DSnoT-Affine | 52.20 | 64.35 | **58.28** | 57.79 | 29.70 | 51.38 | 20.05 | 37.12 | **39.21** |

Table 2: Comparison of perplexity and zero-shot dataset accuracy between joint optimization and sequential optimization under different compression configurations on LLAMA3.

| Model | Bits | Sparsity | Method | PPL ↓ | | | Accuracy (%) ↑ | | | | | |
| | | | | WikiText2 | C4 | Avg. | PIQA | HellaSwag | WinoGrande | ARC-c | ARC-e | Avg. |
|---|---|---|---|---|---|---|---|---|---|---|---|---|
| LLaMA3-8B | w16a16 | − | − | 6.04 | 8.88 | 7.46 | 79.70 | 60.17 | 72.69 | 50.42 | 80.09 | 68.61 |
| | w4a16g128 | 50% | Sequential-Wanda | 10.89 | 15.65 | 13.27 | 74.21 | 50.76 | 67.79 | 35.49 | 67.97 | 59.24 |
| | w4a16g128 | 50% | Joint-Wanda | 9.13 | 13.62 | **11.38** | 74.70 | 51.84 | 68.24 | 38.65 | 71.54 | **60.99** |
| | w4a16g128 | 50% | Sequential-DSnoT | 10.06 | 14.87 | 12.46 | 74.53 | 50.97 | 67.24 | 38.22 | 70.70 | 60.33 |
| | w4a16g128 | 50% | Joint-DSnoT | 9.01 | 13.58 | **11.30** | 74.75 | 51.85 | 68.03 | 38.48 | 71.33 | 60.89 |
| | w3a16g128 | 50% | Sequential-Wanda | 15.23 | 22.10 | 18.66 | 68.44 | 60.61 | 43.07 | 28.66 | 60.60 | 52.28 |
| | w3a16g128 | 50% | Joint-Wanda | 10.78 | 15.72 | **13.25** | 71.54 | 65.03 | 48.73 | 33.61 | 65.40 | **56.86** |
| | w3a16g128 | 50% | Sequential-DSnoT | 15.51 | 22.56 | 19.04 | 68.28 | 43.36 | 61.24 | 28.58 | 61.74 | 52.64 |
| | w3a16g128 | 50% | Joint-DSnoT | 10.66 | 15.73 | **13.20** | 72.47 | 48.49 | 64.40 | 34.30 | 65.78 | **57.09** |

BESA (Xu et al., 2024) for combined compression. For fair comparison, we adopt per-channel weight quantization, per-token activation quantization, and unstructured sparsity across all experiments.

**Evaluation.** To evaluate the efficacy of the LLM compression achieved through the joint optimization strategy, we conducted evaluations across a range of benchmark zero-shot accuracies, encompassing PIQA (Bisk et al., 2020), BoolQ (Clark et al., 2019), ARC (Clark et al., 2018), HellaSwag (Zellers et al., 2019), and WinoGrande (Sakaguchi et al., 2019). Leveraging the lm-eval-harness (Gao et al., 2021), we executed all zero-shot tasks and reported precision results for each benchmark, as well as the overall average accuracy. Furthermore, we conducted perplexity evaluations on the WikiText2 (Merity et al., 2016) and C4 (Raffel et al., 2020) datasets, serving as stable and robust indicators of model generation performance.

## 4.2 RESULTS

As shown in Tabs. 1, 2, 3 and 5 we evaluate the perplexity and zero-shot task accuracy of joint optimization and sequential optimization under different compression configurations using LLaMA1, LLaMA2, LLaMA3, and OPT models of various scales. The experimental results demonstrate

Table 3: Comparison of perplexity and five zero-shot datasets accuracy between sequential optimization and joint optimization on LLaMA1&2.

| Model | Bits | Sparsity | Method | PPL ↓ | | | Accuracy(%) ↑ | | | | | |
|---|---|---|---|---|---|---|---|---|---|---|---|---|
| | | | | WikiText2 | C4 | Avg. | BoolQ | HellaSwag | WinoGrande | ARC-c | ARC-e | Avg. |
| | w16a16 | – | – | 4.88 | 6.73 | 5.81 | 80.55 | 60.04 | 72.21 | 48.46 | 79.37 | 68.13 |
| | w2a16g128 | 50% | Sequential-Wanda | 10.88 | 13.63 | 12.26 | 61.83 | 44.32 | 56.59 | 28.58 | 59.55 | 50.17 |
| | w2a16g128 | 50% | Joint-Wanda-Affine | 9.17 | 12.26 | **10.72** | 66.20 | 45.00 | 58.01 | 31.48 | 62.83 | **52.70** |
| LLaMA2-13B | w2a16g128 | 50% | Sequential-DSnoT | 11.83 | 14.49 | 13.16 | 59.51 | 43.58 | 58.16 | 29.60 | 58.33 | 49.84 |
| | w2a16g128 | 50% | Joint-DSnoT-Affine | 9.44 | 12.52 | **10.98** | 64.37 | 43.95 | 57.69 | 28.58 | 57.44 | **50.41** |
| | w2a16g128 | 60% | Sequential-BESA | 12.50 | 14.80 | 13.65 | 63.18 | 42.76 | 56.69 | 27.30 | 58.12 | 49.61 |
| | w2a16g128 | 60% | Joint-BESA-Affine | 10.75 | 13.70 | **12.23** | 67.09 | 42.69 | 55.80 | 29.94 | 59.09 | 50.92 |
| | w16a16 | – | – | 5.63 | 7.07 | 6.35 | 75.10 | 56.95 | 69.85 | 41.89 | 75.29 | 63.82 |
| | w2a16g128 | 50% | Sequential-Wanda | 12.68 | 16.29 | 14.49 | 48.16 | 39.47 | 53.98 | 24.82 | 52.39 | 43.76 |
| | w2a16g128 | 50% | Joint-Wanda-Affine | 10.89 | 14.48 | **12.69** | 62.01 | 40.40 | 55.16 | 25.85 | 55.59 | **47.80** |
| LLaMA-7B | w2a16g128 | 50% | Sequential-DSnoT | 13.32 | 16.98 | 15.15 | 40.45 | 38.96 | 53.82 | 24.14 | 52.81 | 42.04 |
| | w2a16g128 | 50% | Joint-DSnoT-Affine | 11.84 | 15.60 | **13.72** | 57.64 | 41.04 | 56.35 | 27.98 | 58.03 | **48.21** |
| | w2a16g128 | 60% | Sequential-BESA | 16.05 | 20.03 | 18.04 | 49.26 | 36.85 | 51.85 | 24.14 | 48.14 | 42.05 |
| | w2a16g128 | 60% | Joint-BESA-Affine | 14.59 | 19.17 | **16.88** | 53.48 | 37.24 | 55.48 | 23.72 | 50.37 | **44.06** |
| | w16a16 | – | – | 5.03 | 6.61 | 5.82 | 77.98 | 59.91 | 72.77 | 46.50 | 77.35 | 66.90 |
| | w2a16g128 | 50% | Sequential-Wanda | 9.17 | 11.79 | 10.48 | 62.93 | 45.68 | 61.40 | 29.01 | 60.26 | 51.86 |
| | w2a16g128 | 50% | Joint-Wanda-Affine | 8.17 | 10.98 | **9.58** | 65.96 | 46.17 | 63.69 | 31.31 | 66.03 | **54.63** |
| LLaMA-13B | w2a16g128 | 50% | Sequential-DSnoT | 9.52 | 12.13 | 10.83 | 64.18 | 45.48 | 61.79 | 27.90 | 60.35 | 51.94 |
| | w2a16g128 | 50% | Joint-DSnoT-Affine | 8.92 | 11.63 | **10.28** | 66.11 | 46.45 | 63.14 | 31.65 | 64.30 | **54.33** |
| | w2a16g128 | 60% | Sequential-BESA | 11.23 | 13.72 | 12.48 | 63.42 | 43.18 | 58.95 | 27.38 | 57.23 | 50.03 |
| | w2a16g128 | 60% | Joint-BESA-Affine | 10.41 | 13.61 | **12.01** | 63.39 | 44.57 | 61.08 | 29.69 | 62.12 | **52.17** |

Table 4: The contribution of the reordering method to the combined compression approach based on learned sparsification masks.

| Model | Bits | Sparsity | Method | PPL ↓ | | | Accuracy (%) ↑ | | | | | |
|---|---|---|---|---|---|---|---|---|---|---|---|---|
| | | | | WikiText2 | C4 | Avg. | BoolQ | HellaSwag | WinoGrande | ARC-c | ARC-e | Avg. |
| | w16a16 | – | – | 5.63 | 7.07 | 6.35 | 75.10 | 56.95 | 69.85 | 41.89 | 75.29 | 63.82 |
| LLaMA-7B | w2a16g128 | 50% | Joint without Reorder | 114.72 | 316.01 | 215.36 | 37.82 | 25.87 | 50.83 | 19.19 | 27.78 | 32.30 |
| | w2a16g128 | 50% | Joint with Reorder | 13.85 | 19.06 | **16.46** | 58.68 | 38.34 | 52.80 | 24.57 | 52.44 | **45.37** |
| | w16a16 | – | – | 4.88 | 6.73 | 5.81 | 80.55 | 60.04 | 72.21 | 48.46 | 79.37 | 68.13 |
| LLaMA2-13B | w2a16g128 | 50% | Joint without Reorder | 236.09 | 586.26 | 411.18 | 37.88 | 25.93 | 50.19 | 19.96 | 26.59 | 32.11 |
| | w2a16g128 | 50% | Joint with Reorder | 11.88 | 14.99 | **13.44** | 59.26 | 40.82 | 52.72 | 25.08 | 52.18 | **46.12** |

that joint optimization consistently exhibits performance improvements across various combinations of compression methods (for example, LLaMA3-8B with w3a16g128 and 50% sparsity achieved 56.86% accuracy vs. 52.28% with sequential optimization on zero-shot datasets). Notably, in scenarios with high compression rates, the performance gains from joint optimization are even more significant. Specifically, on the LLaMA-7B model using the w2a16g128 quantization configuration with 75% sparsity and employing the Wanda (Sun et al., 2023) sparsification method, joint optimization reduces the perplexity on the WikiText2 dataset by 3476.24 compared to sequential optimization (44.79 vs. 3521.03). These experiments further substantiate the substantial benefits of joint optimization over sequential optimization.

Tab. 6 presents the results of combined compression for larger-scale models of LLaMA 1 and 2. The experiments indicate that our method continues to outperform the latest combined compression techniques. For instance, with a quantization configuration of w2a16g128 and 75% sparsity for LLaMA-30B, the joint optimization strategy using the Wanda sparsity method reduced the average perplexity on the WikiText2 and C4 datasets by 519.46 (from 577.02 to 57.56). Similarly, for LLaMA2-70B with the same quantization configuration and sparsity rate, the joint optimization strategy using the DSnoT sparsity method achieved a reduction in average perplexity by 79.41 (from 102.67 to 23.26).

### 4.3 ABLATION STUDY

**Reordering Effectiveness.** As shown in Tab. 4, we demonstrate the effectiveness of the reordering method for learnable sparse masks on dynamic weights. On the LLaMA2-13B model with the

Table 5: Comparison of perplexity and zero-shot dataset accuracy between sequential optimization and joint optimization on OPT models of different scales.

| Model | Bits | Sparsity | Method | PPL ↓ | | | Accuracy (%) ↑ | | | | | |
| | | | | WikiText2 | C4 | Avg. | PIQA | HellaSwag | WinoGrande | ARC-c | ARC-e | Avg. |
|---|---|---|---|---|---|---|---|---|---|---|---|---|
| OPT-125M | w16a16 | – | – | 26.86 | 24.60 | 25.73 | 62.89 | 29.19 | 50.43 | 19.02 | 43.56 | 41.02 |
| | w2a16g128 | 50% | Sequential-Wanda | 425.31 | 493.04 | 459.18 | 54.05 | 25.82 | 49.64 | 20.56 | 30.59 | 36.13 |
| | w2a16g128 | 50% | Joint-Wanda-Affine | 98.47 | 119.93 | **109.20** | 55.44 | 26.72 | 51.93 | 17.06 | 35.77 | **37.38** |
| | w2a16g128 | 50% | Sequential-DSnoT | 539.87 | 556.32 | 548.10 | 54.08 | 25.84 | 50.43 | 18.77 | 29.12 | 35.65 |
| | w2a16g128 | 50% | Joint-DSnoT-Affine | 106.75 | 119.68 | **113.22** | 56.85 | 26.81 | 50.11 | 17.06 | 34.17 | **37.00** |
| OPT-1.3B | w16a16 | – | – | 14.27 | 14.72 | 14.50 | 71.59 | 41.48 | 59.90 | 23.37 | 56.90 | 50.65 |
| | w2a16g128 | 50% | Sequential-Wanda | 42.25 | 51.81 | 47.03 | 60.66 | 31.44 | 54.93 | 20.30 | 42.04 | 41.87 |
| | w2a16g128 | 50% | Joint-Wanda-Affine | 28.65 | 41.19 | **34.92** | 60.60 | 31.69 | 53.03 | 20.98 | 43.39 | **41.94** |
| | w2a16g128 | 50% | Sequential-DSnoT | 48.25 | 57.53 | 52.89 | 60.39 | 30.66 | 51.85 | 20.81 | 40.78 | 40.90 |
| | w2a16g128 | 50% | Joint-DSnoT-Affine | 29.23 | 41.30 | **35.27** | 60.99 | 31.97 | 53.11 | 21.92 | 43.56 | **42.31** |
| OPT-2.7B | w16a16 | – | – | 12.18 | 13.16 | 12.67 | 73.77 | 45.85 | 60.77 | 26.79 | 60.77 | 53.59 |
| | w2a16g128 | 50% | Sequential-Wanda | 429.37 | 860.91 | 645.14 | 58.05 | 26.68 | 50.67 | 18.00 | 34.34 | 37.55 |
| | w2a16g128 | 50% | Joint-Wanda-Affine | 19.95 | 29.00 | **24.48** | 65.12 | 34.64 | 53.59 | 20.98 | 48.65 | **44.60** |
| | w2a16g128 | 50% | Sequential-DSnoT | 546.27 | 1263.97 | 905.12 | 56.36 | 26.59 | 49.17 | 19.45 | 31.10 | 36.53 |
| | w2a16g128 | 50% | Joint-DSnoT-Affine | 20.28 | 28.83 | **24.56** | 64.85 | 34.58 | 56.11 | 20.05 | 46.92 | **44.50** |
| OPT-6.7B | w16a16 | – | – | 10.63 | 11.74 | 11.19 | 76.27 | 50.50 | 65.19 | 30.46 | 65.57 | 57.60 |
| | w2a16g128 | 50% | Sequential-Wanda | 17.59 | 21.64 | **19.62** | 68.66 | 39.70 | 58.72 | 25.93 | 56.90 | **49.98** |
| | w2a16g128 | 50% | Joint-Wanda-Affine | 15.33 | 23.96 | 19.65 | 68.66 | 39.04 | 58.08 | 22.95 | 55.76 | 48.90 |
| | w2a16g128 | 50% | Sequential-DSnoT | 18.53 | 22.43 | 20.48 | 68.38 | 38.79 | 58.48 | 23.97 | 53.66 | 48.66 |
| | w2a16g128 | 50% | Joint-DSnoT-Affine | 15.24 | 23.18 | **19.21** | 68.28 | 39.43 | 57.22 | 23.72 | 55.38 | **48.81** |
| OPT-13B | w16a16 | – | – | 9.85 | 11.19 | 10.52 | 75.84 | 52.42 | 65.04 | 32.93 | 67.12 | 58.67 |
| | w2a16g128 | 50% | Sequential-Wanda | 19.52 | 24.75 | 22.14 | 67.24 | 40.42 | 60.37 | 26.19 | 52.90 | 49.42 |
| | w2a16g128 | 50% | Joint-Wanda-Affine | 13.86 | 21.27 | **17.57** | 68.38 | 41.94 | 58.72 | 25.08 | 58.16 | **50.46** |
| | w2a16g128 | 50% | Sequential-DSnoT | 19.97 | 24.78 | 22.38 | 68.44 | 39.95 | 61.80 | 25.93 | 51.34 | 49.49 |
| | w2a16g128 | 50% | Joint-DSnoT-Affine | 13.87 | 21.21 | **17.54** | 68.71 | 41.52 | 60.22 | 25.42 | 56.10 | **50.39** |

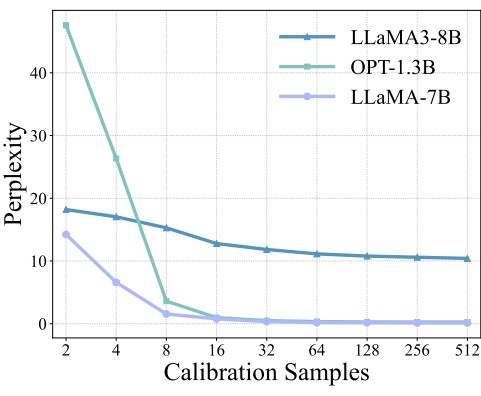 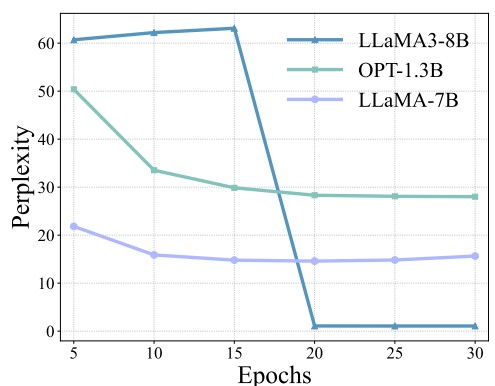

Figure 3: Impact of calibration samples and number of epochs on joint optimization performance. We evaluate the perplexity of WikiText2 across different models using the w2a16g128 quantization configuration with 50% sparsity. To enhance visualization clarity, the perplexities of the OPT model and LLaMA-7B model (left part) are divided by 100, while the perplexity of the LLaMA3 model (right part) is divided by 10.

w2a16g128 quantization configuration and 50% sparsity, the reordering method reduces the average perplexity on WikiText2 and C4 by 397.74 compared to the method without reordering (13.44 vs. 411.18). Thus, the reordering method ensures that less important weight parameters are sparsified in each iteration, thereby improving the performance of the compressed model.

**Impact of Calibration Data and Epochs.** Fig. 3 illustrates the impact of varying amounts of calibration data and epochs on the performance of joint optimization. Overall, as the number of samples and epochs increases, perplexity generally decreases across different models. This trend indicates that more calibration data and training epochs are beneficial for joint optimization. Additionally, the

Table 6: Comparison of perplexity between joint and sequential optimization of larger scale models on LLAMA1&2.

| Model | Bits | Sparsity | Method | PPL ↓ | | |
| --- | --- | --- | --- | --- | --- | --- |
| | | | | WikiText2 | C4 | Avg. |
| LLaMA-30B | w16a16 | − | − | 4.04 | 5.97 | 5.01 |
| | w2a16g128 | 75% | Sequential-Wanda | 392.89 | 761.15 | 577.02 |
| | w2a16g128 | 75% | Joint-Wanda | 43.39 | 71.74 | **57.56** |
| | w2a16g128 | 75% | Sequential-DSnoT | 443.38 | 967.83 | 705.60 |
| | w2a16g128 | 75% | Joint-DSnoT | 44.67 | 73.44 | **59.05** |
| LLaMA-2-70B | w16a16 | − | − | 3.32 | 5.52 | 4.42 |
| | w2a16g128 | 75% | Sequential-Wanda | 81.99 | 75.85 | 78.92 |
| | w2a16g128 | 75% | Joint-Wanda | 17.18 | 22.11 | **19.65** |
| | w2a16g128 | 75% | Sequential-DSnoT | 105.35 | 99.99 | 102.67 |
| | w2a16g128 | 75% | Joint-DSnoT | 18.42 | 28.10 | **23.26** |

OPT and LLaMA3 models show greater sensitivity to changes in the number of calibration samples and epochs, respectively.

**Order of compression methods.** Fig. 4 in the appendix illustrates why joint optimization does not apply sparsification first ($\mathcal{Q}(AM \odot W)$). The $zp$ introduced in Eq. 1 maps the zero elements of the sparse matrix $MW$ to non-zero values, significantly reducing the sparsity. Therefore, to achieve the target sparsity rate, we adhere to the approach defined in Eq. 7 for joint optimization of combined compression.

## 5 CONCLUSION

Post-training quantization and sparsification methods have shown considerable promise in the compression of large language models. However, previous combined compression approaches have relied on sequential optimization, separately minimizing quantization and sparsification errors. This practice has led to a significant increase in mean squared error (MSE) loss, especially in high compression rate configurations. Our combined compression approach mitigates this issue by simultaneously optimizing both quantization and sparsification errors. Additionally, learnable sparsification masks often fail to appropriately sparsify less important weights when dealing with dynamic weights. To overcome this, we propose a reordering method that prioritizes the sparsification of lower-importance weights in each iteration. This approach ensures stability and convergence in the optimization process, further reducing the MSE of the objective function. Our method consistently improves performance across various compression configurations and models. Notably, the combined compression method exhibits substantial potential for enhancing model performance, particularly in high compression rate scenarios. Future research could focus on developing more effective learning strategies for joint optimization to further advance this field.

## 6 LIMITATIONS

Due to limited resources, the experiments in this study were primarily conducted on a selected subset of natural language processing datasets and models. Conducting experiments on a broader range of datasets and models would provide a more comprehensive demonstration of the method's generalization and robustness.

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

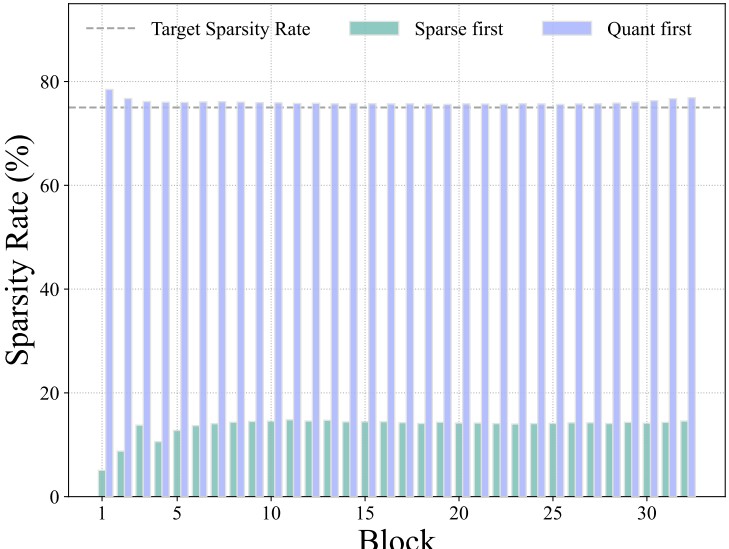

Figure 4: Comparison of sparsity rates using different sequential compression methods for various blocks. The method of quantizing first and then sparsifying achieves the target sparsity rate. In contrast, performing sparsification first can be undermined by the zero points introduced during quantization, leading to a final sparsity rate significantly lower than the intended target.

## A  APPENDIX

### A.1  BIAS COMPENSATING

Given that we use only a small amount of data for compression optimization, avoiding model overfitting presents a significant challenge. To mitigate this, we aim to approximate the final output of the compressed model to match that of the original model. By ensuring similar output distributions, we can better preserve the model's generalization ability. Consequently, it is necessary to compensate for the bias in the linear layer. Specifically, we proceed with the following derivation:

$$Y = XW + bias \tag{10}$$
$$= XA^{-1}AW + bias, \tag{11}$$
$$= (X - \delta)A^{-1}AW + (bias + \delta W), \tag{12}$$
$$\approx (X - \delta)A^{-1}\mathcal{Q}(AW) + (bias + \delta W), \tag{13}$$
$$\approx (X - \delta)A^{-1}(M \odot \mathcal{Q}(AW)) + (bias + \delta W). \tag{14}$$

Ultimately, we need to compensate for the $bias$ with $\delta W$ to maintain approximate output equivalence. Consistent with previous research (Shao et al., 2023; Ma et al., 2024; Xu et al., 2024), we approximate $\mathcal{Q}(AW) \approx AW$ and $M \odot \mathcal{Q}(AW) \approx \mathcal{Q}(AW)$ to ensure equivalent model outputs.

