# OpenReview forum: "Extreme composite compression of large language models through joint optimization"
_ICLR.cc/2025/Conference — Submitted to ICLR 2025_

### Official Review · Reviewer_DcjF · 2024-10-28

**Soundness:** 3
**Presentation:** 2
**Contribution:** 3
**Rating:** 6
**Confidence:** 3

**Summary:**

This paper presents a novel approach to the joint optimization of quantization and sparsity in large language models (LLMs).
The paper's main contribution lies in proposing a new joint optimization strategy that can simultaneously minimize errors from both quantization and sparsity, particularly suitable for compressing large language models in low-bit and high-sparsity configurations. Additionally, the paper introduces a dynamic reordering method to enhance the effectiveness of learnable masks, bringing significant performance improvements to the field of model compression.

**Strengths:**

This paper has the following strengths:
(1) The paper presents a novel joint optimization strategy for quantization and sparsification of large language models. This approach is innovative as it addresses the issue of error amplification in traditional methods where quantization precedes sparsification.
(2) The research demonstrates high quality through its comprehensive experimental design and rigorous evaluation.
(3) This work uses  a dynamic reordering method to enhance the effectiveness of learnable masks.

**Weaknesses:**

This paper has two shortcomings, as follows:
（1）The paper could benefit from a stronger theoretical grounding of why joint optimization works better than sequential approaches.
（2）The paper lacks a clear explanation of some of the conclusions and mathematical formulas.
For example, line 78 describes "Our experiments indicate that initiating with quantization optimization followed by the
application of weight sparsity amplifies the quantization errors, consequently increasing the overall
mean squared error loss",  but there is no corresponding experimental result to support this conclusion. “

**Questions:**

Relevant questions and suggestions are given below
(1) What is the calculation of  $||.||^2_{F}$ in equations 4,5, and 6? It is advisable to give clarification in the paper.
(2) How the inference speed or throughput of the model is affected after quantization and sparsification on a specific GPU?

---

### Official Review · Reviewer_4cJU · 2024-10-31

**Soundness:** 3
**Presentation:** 3
**Contribution:** 2
**Rating:** 3
**Confidence:** 4

**Summary:**

The paper presents a method to jointly optimize for quantization and sparsity, unlike previous methods that alternate between optimizing the quantization loss and the sparsity loss. Joint optimization improves substantially over alternating optimization, across bit-widths and sparsity levels.

**Strengths:**

The proposed methods gives substantial gains over sequential optimization for int2 and int3 quantization with 50%-75% sparsity.

**Weaknesses:**

- The paper does not provide results for latency. The authors must give latency numbers all the settings presented in the paper. This is to get a better understanding of latency vs quality tradeoff.
- Although the results presented in the paper are impressive, the paper does not add significant contributions on top of existing works.

**Questions:**

- How do these methods perform with lower sparsity levels? Let's say for 5%, 10%, 15% and 20% sparsity. From a practicality point of view, this seems to be the more ideal setting, since the results in the paper for Quantization + Sparsity are very far from the bf16 numbers in most settings.
- Have the authors tried integrating other post-training quantization methods such as GPTQ, or QuIP [1] and FrameQuant [2] for int2 quantization?
- For the same number of parameters, is quantization + sparsity better than only quantization or only sparsity?

Chee, Jerry et al. “QuIP: 2-Bit Quantization of Large Language Models With Guarantees.” NeurIPS 2023.

Adepu, Harshavardhan et al. “FrameQuant: Flexible Low-Bit Quantization for Transformers.” ICML 2024.

---

### Official Review · Reviewer_Ktof · 2024-11-02

**Soundness:** 2
**Presentation:** 2
**Contribution:** 1
**Rating:** 3
**Confidence:** 4

**Summary:**

This paper proposes a joint optimization strategy for compressing large language models (LLMs) through post-training quantization and sparsification, integrating both processes to minimize errors simultaneously. Experiments demonstrate performance improvements on LLaMA and OPT models over sequential approach.

**Strengths:**

- **Integrated Strategy**: The proposed training strategy effectively links quantization and sparsification, aiming to improve on previous sequential approaches.
- **Broad Compatibility**: The approach is compatible with various quantization and sparsification techniques, making it adaptable across different LLM architectures.

**Weaknesses:**

- **Limited Novelty**: The compression strategy primarily combines existing techniques (i.e., AffineQuant and DSnoT) to compress LLMs without introducing something novel and substantial.
- **Unclear Motivation for Reordering**: The purpose and intuition behind the reordering mechanism are not clearly explained, especially regarding the design of the importance metric in Equation 9. Details on how reordering is applied during training are also missing.
- **Marginal Improvement over Sequential Approaches**: The performance gains over sequential methods are limited, and the ablation study raises questions about the joint optimization’s effectiveness.
	- Results in Tables 3 and 4 indicate that the reordering mechanism is critical for the success of joint optimization, which is not sufficiently emphasized. For instance, in the LLaMA2-13B model, "Joint without reorder" (PPL 411.18) performs significantly worse than the baseline “Sequential-Wanda” (PPL 13.56), suggesting that joint optimization alone may not avoid local optima as claimed. This inconsistency challenges the rationale of the method.
- **Lack of Clarity and Visual Presentation**: The paper’s layout is inconsistent, with texts in Figure 3 and Table 6 much larger than in other tables and figures. Additionally, there is no visual demonstration of the proposed method to aid understanding.
- **Limited Task Diversity**: Evaluation is limited to standard zero-shot NLP tasks. Testing on a wider range of tasks and datasets would provide a stronger case for general applicability.

**Questions:**

- What are the exact operations and steps for using the importance weights computed in Equation 9 to reorder quantized weights?
- In line 316, why is the sparsity ratio $\beta$ written as $5e^0$ instead of simply $5$? Is this a typo?
- Could using more advanced quantization and sparsification methods yield better results?

---

### Official Review · Reviewer_kMbv · 2024-11-04

**Soundness:** 3
**Presentation:** 3
**Contribution:** 2
**Rating:** 5
**Confidence:** 4

**Summary:**

There are two well-known approaches towards post-training LLM compression: quantization and sparsification. One can apply both techniques at the same time to compress the LLM. Traditionally, these techniques are applied sequentially, which can lead to significant accuracy losses due to compounding errors. The authors  assert that simultaneous optimization of both quantization and sparsification errors can enhance performance by mitigating alignment issues that arise in sequential processes. They propose a learnable transformation matrix and a reordering method within the sparsification process to improve weight selection stability. The proposed method is tested across various LLM backbones and compression configurations, demonstrating superior benchmark accuracy and efficiency compared to sequential methods, especially under high-compression scenarios with low-bit quantization and high sparsity rates.

**Strengths:**

1. The paper appropriately points out the sequential application of quantization and sparsification results in sub-optimal results.

2. Experiments show that joint optimization can better recover the model accuracy compared to other approaches.

**Weaknesses:**

1. The method only assessed in terms of easy benchmarks such as common sense reasoning. Comparisons on important benchmarks such as MMLU, GPQA, GSM8K, etc. would benefit the paper.

2. For unstructured sparsity, the weight masks must also be stored and utilized at inference time. This results in additional storage and latency overhead compared to dense quantization. In this sense, a trade-off analysis regarding memory usage and accuracy is needed.

**Questions:**

Please refer to the Weaknesses section.

---

### Meta-Review · Area_Chair_v2Jh · 2024-12-14

**Metareview:**

This paper proposes a joint optimization strategy for compressing LLMs through post-training quantization and sparsification, integrating both processes to minimize errors simultaneously. It received scores of 3356. Several major concerns still remain, including (1) insufficient evaluation, comparisons on important benchmarks such as MMLU, GPQA, GSM8K are needed, (2) more analysis on latency, trade-off between memory usage and accuracy is needed, (3) limited novelty and unclear motivation for reordering, and (4) the performance gains over sequential methods are limited. Overall, the AC would like to recommend rejection of the paper.

**Additional Comments On Reviewer Discussion:**

The reviewers have asked many questions in the review comments; however, no rebuttal is provided, so all the concerns remain.

---

### Decision · Program_Chairs · 2025-01-22

Reject